

# Development of a travel recommendation algorithm based on multi-modal and multi-vector data mining

Ruixiang Liu

Nanchang Normal University, Nanchang, China

## ABSTRACT

Given the rise of the tourism industry, there is an increasing urgency among tourists to access information about various tourist attractions. To address this challenge, innovative solutions have emerged, utilizing recommendation algorithms to offer customers personalized product recommendations. Nonetheless, existing recommendation algorithms predominantly rely on textual data, which is insufficient to harness the full potential of online tourism data. The most valuable tourism information is often found in the multi-modal data on social media, characterized by its voluminous and content-rich nature. Against this backdrop, our article posits a groundbreaking travel recommendation algorithm that leverages multi-modal data mining techniques. The proposed algorithm uses a travel recommendation platform, designed using multi-vector word sense segmentation and multi-modal data fusion, to improve the recommendation performance by introducing topic words. In our final experimental comparison, we verify the recommendation performance of the proposed algorithm on the real data set of TripAdvisor. Our proposed algorithm has the best degree of confusion with various topics. With a LOP of 20, the Precision and MAP values reach 0.0026 and 0.0089, respectively. It has the potential to better serve the tourism industry in terms of tourist destination recommendations. It can effectively mine the multi-modal data of the tourism industry to generate more excellent economic and social value.

## INTRODUCTION

The advent of the internet, cloud computing, and mobile electronic devices have engendered a gradual decline in computing and storage costs, accompanied by an unprecedented surge in the accumulation of electronic data across diverse fields. The dissemination of internet data is no longer confined to text-based content. Still, it encompasses multiple modalities, including images, audio, and videos, giving rise to a multi-modal data landscape. Given that the internet is fundamentally rooted in sharing, interaction, virtualization, and service, all intimately intertwined with social functions, the underlying foundation of the internet is the unimpeded flow of data. The internet facilitates the free flow of data between individuals and devices, connecting the world through data. This revolutionary transformation facilitated

Corresponding author
Ruixiang Liu,
liuruixiang@ncnu.edu.cn

by the internet has enabled humans to connect, share information, explore their interests, and mutually benefit from these interactions.

Intelligent recommendations are facilitated by data sharing. Research scholars have proposed personalized recommendations to address user interest discovery and intelligent recommendations in internet communication (*Zhou, Li & Liang, 2020*). A customized recommendation is a service that employs intelligent algorithms to analyze data such as users' historical information and automatically recommend content or products of interest to them. However, most recommendation algorithms currently in use rely solely on textual data. In contrast, the largest volume of data, the richest content, and the most valuable information available online are found in the multi-modal data in social media. Social media data comprises images, text, videos, audio, and other modalities, which possess rich content and exhibit a certain correlation between different modalities. Therefore, if multi-modal data can be effectively utilized and mined, it can generate more significant economic benefits and social value. Especially in the tourism industry, where data is generated and consumed by humans, multi-modal data often contains interesting content or useful knowledge. This is why the usability and effectiveness of data mining are widely researched and recognized in academia and the tourism industry. Social media platforms generate much text and image data about users' travel. By mining multi-modal data from these platforms, we can better understand users' travel interests or intentions and play a critical role in the semantic expression and fusion of multi-modal data.

Currently, text and image are the two most critical forms of multi-modal data. Their corresponding data mining analyses have been extensively studied, achieving remarkable results in various application areas (*Kruk et al., 2019*). However, compared to traditional structured data, text and image data are characterized by their vast volume and diversity. These characteristics pose new challenges for computer data processing and automated data understanding, especially for tourism data that is diverse and high-dimensional. With the exponential growth of text data, the processing speed of a single process for text data can no longer meet practical application needs (*De Neys, 2021*). Similarly, images exhibit characteristics such as multi-category, multi-instance, and multi-scene, increasing the complexity of their contents.

Given this context, the processing and analysis of vast and exponentially growing multi-modal data present significant challenges for traditional text and image mining analysis methods regarding efficiency and accuracy. Consequently, researchers have devoted considerable attention to exploring effective mining and analyzing large-scale multi-modal data techniques. In this article, we leverage travel social media user behaviour data as a starting point and employ deep learning statistical techniques to analyze text and image data. Our approach enables the computer to comprehend data content and extract confidential information, including the internal relationships between data, through multi-vector semantic segmentation. Additionally, our proposed topic and multi-modal sentiment perception models effectively mine tourism data in unsupervised scenarios. Specifically, our model can autonomously identify topics and corresponding sentiment information from text or multi-modal data and utilize a sentiment dictionary to determine

the sentiment polarity of each topic. We also integrate cross-domain data into our model to facilitate travel recommendation tasks and design a platform accordingly.

## RELATED WORKS

The fundamental concept of recommendation systems entails utilizing specific information filtering methods to suggest various items to customers who may find them appealing. In essence, recommendation systems leverage users' historical behaviours, interest points, and other relevant information to determine their current needs or preferences (*Das, Sahoo & Datta, 2017*). With the advancement of information technology, data in many application domains has significantly increased in the number of samples and feature dimensions, resulting in higher computational costs for data classification, clustering, and retrieval processes. To mitigate these computational expenses effectively and account for data sparsity on tourism websites, content-based recommendation algorithms have emerged as a more prominent solution for personalized tourism recommendations.

Typical content-based recommendation methods generally focus solely on user features, disregarding the sentiment information present in user data. In recognition of this issue, *Lops et al. (2019)* highlights that reviews contain various pieces of information that can mitigate the problem of sparsity. As a result, these works extract the sentiment associated with user features from user reviews and utilize it for a recommendation. *Salehan & Kim (2016)* proposes a recommendation algorithm that mines customer review content from a product web page and uses the data to calculate scores, dynamically ranking web pages based on those scores. Subsequently, research scholars constructed user interest models based on recommendation algorithms, with *Javed et al. (2021)* utilizing a content-based approach to construct user interest models by extracting text features. Although the content-based approach is highly interpretable and offers easy-to-understand recommendation reasons without the cold-start problem, it lacks recommendation diversity. It proves challenging to explore users' potential interests. *Tewari (2020)* tackles this problem with a partitioned hybrid recommendation approach, where collaborative filtering and content-based methods generate recommendation lists for users to select content. While this hybrid approach mitigates the diversity deficit, it brings the cold-start problem (*Li et al., 2020*). In response, *Allan et al. (2003)* proposes that user interest modelling is, in fact, a method that can provide users with personalized information services, serving as a core component of an information retrieval or filtering system that can capture each user's unique information needs. *Zheng et al. (2010)* argues that user preferences for information can be calculated by summarising their browsing behaviour, browsing content, and knowledge background, *i.e.,* the user interest model. *Wei & Li (2019)* constructs a document filter based on user behaviour rules to assist users in finding exciting content. In contrast, *Liu et al. (2022)* proposes that a user model is a computable description of users' interests, preferences, and patterns, constituting a personalized service.

The foregoing interest models were posited as the foundational framework for travel recommendation algorithms. Some scholars contend that the user's interest model can be valuable for extracting sentiment information from reviews and augmenting

recommendations (*Zheng, Burke & Mobasher, 2012*). This approach addresses the challenges of cold start, data sparsity, and multi-modal data application in travel recommendation. However, such studies overlook the thematic information of attractions solely from the user's perspective. The approach for modelling the user's interest warrants scrutiny.

In *Chen et al. (2021)*, personalized similarity (PAS) modelling was designed to generate recommendations incorporating heterogeneous tourism information. This approach considers attraction themes' sentiment and uses multi-modal data but falls short in utilizing ratings from tourists who have visited these attractions. While some recommendation methods consider emotional factors, they rely on textual information alone and neglect the multi-modal dimension of travel media data. Additionally, they fail to consider multi-modal review information's role in assessing attractions. Various techniques have been developed to handle the obstacles faced in multi-modal data mining in practical applications, including cross-modal data mining and word sense segmentation (*Wang et al., 2018*).

As an example, a study has demonstrated the use of semantic-based cross-modal data mining, which employs deep convolutional neural networks to enhance cross-modal retrieval (*Qu et al., 2015*). In addition, a multi-modal joint sentiment topic model was introduced to conduct weakly supervised sentiment analysis of microblog texts and emoticons (*Park & Kim, 2021*). Further, a cross-modal retrieval method was proposed in the literature based on subspace learning of multi-sequence discriminative structures to optimize cross-modal retrieval performance (*Fedorov et al., 2022*). Subsequently, scholars have explored topic models based on multi-modal data, such as the introduction of a multi-modal viewpoint topic model, which captures the correlation between textual and visual modalities by integrating user-generated images and text documents (*Fazio, Gallagher & De Klerk, 2022*).

## METHODOLOGY

Modality refers to how information is represented and exchanged on a specific medium. In tourism, multi-modal data consists of text, static images of attractions, and filmed tourism videos. Previous research indicates that multi-modal data harbours more diverse and valuable content than single-modal data, which can enhance the efficacy of data processing. In this article, we present a travel recommendation algorithm founded on multi-modal data mining and multi-vector word sense segmentation technology to attain intelligent recommendation information for users on travel platforms and enhance the precision and efficiency of recommendations.

### Overall design

Figure 1 depicts the framework diagram of the multi-modal sentiment-aware tourism recommendation algorithm, which analyses text and image information on the tourism platform under the model framework diagram. Multi-modal topic mining and sentiment analysis are conducted using data from the tourism domain. After inputting a multi-modal
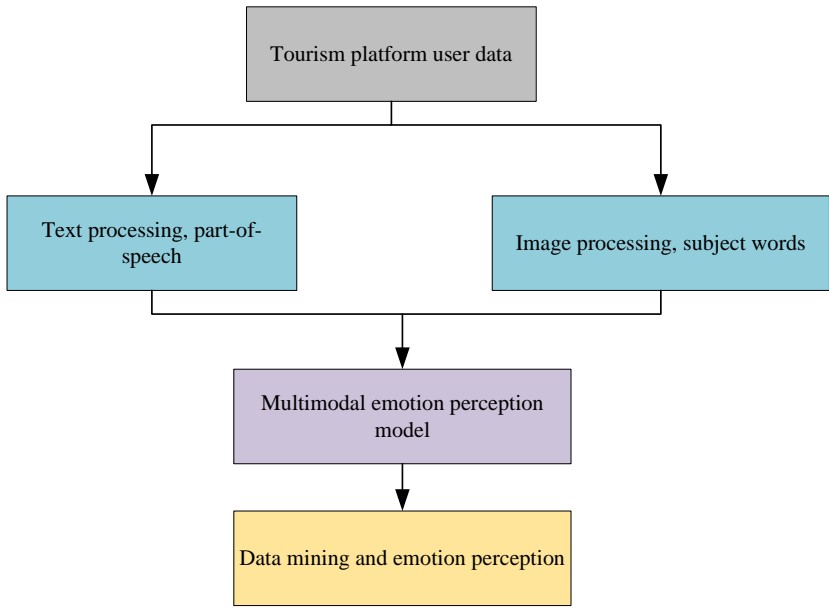

**Figure 1  Flow chart of travel recommendation algorithm based on multi-modal data mining.**

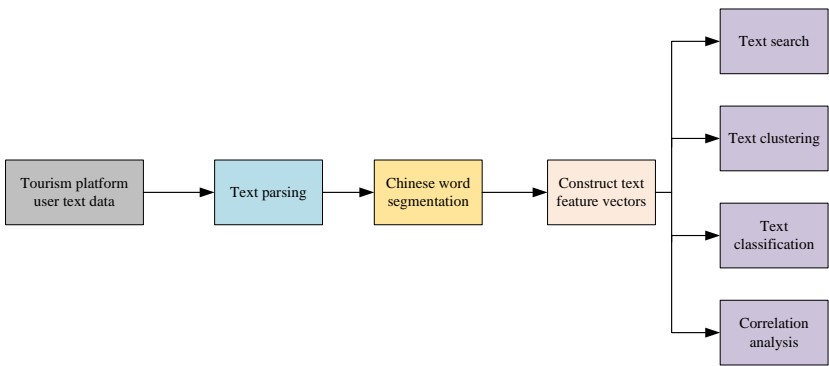

**Figure 2  Text mining analysis flow chart.**

corpus of tourists, the algorithm separately outputs multi-modal topics, including each tourist/attraction and their corresponding sentiment information.

Figure 2 illustrates the flowchart of the mining and analysis of text data in multi-modal data. The text is first parsed, and then the multi-vector word meanings are segmented according to the Chinese sub-word. Text feature vectors are subsequently constructed to complete text retrieval, clustering, classification, and association analysis.

## Feature selection

Feature selection is a critical technique that directly impacts classification accuracy. Its primary purposes are to improve classifier efficiency by reducing the number of adequate

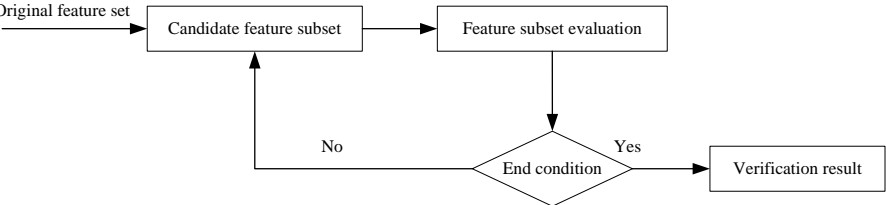

**Figure 3  Feature selection flow chart.**

words and to enhance classification accuracy by eliminating noisy features. As such, feature selection is one of the most effective technical methods for reducing feature dimensionality. By selecting relevant information features from a high-dimensional space, the learning process is accelerated, pattern generalization is improved, the running time of fundamental applications is reduced, and it plays a crucial role in many subject area applications. The feature selection process in this article is depicted in Fig. 3, where the construction of candidate feature subsets and evaluation are performed based on the original feature data set, the performance of the subset that satisfies the conditions is selected, and finally, the validation is concluded. Feature selection enables more effective and accurate data classification by eliminating irrelevant data during data dimensionality reduction, improving learning accuracy, and improving result comprehensibility.

## Multi-modal data mining for tourist users

Multi-modal data of visitors refers to the combination of textual and visual data, where each visitor document contains text and image data. Textual data includes words and sentiment words generated based on the corresponding topic's document-topic distribution and document-sentiment word distribution. Visual data includes visual words generated based on the document-topic distribution. The multi-modal data of visitors is used for topic modelling and sentiment analysis to identify each visitor's preferences.

$$W_t \in 1, 2, 4, \ldots, w^u \tag{1}$$

$$V_t \in 1, 2, 4, \ldots, w^v \tag{2}$$

where $W_t$ is the text theme and $V_t$ is the visual theme. Under the definition of each theme $W_t$, the polynomial distribution on the text theme and visual theme can be obtained according to $\beta_1^u$ and $\beta_2^u$.

$$\varphi^u \in Dir(\beta_1^u) \tag{3}$$

$$\psi^u \in Dir(\beta_2^u) \tag{4}$$

For each visitor topic $W_t \in 1, 2, 4, \ldots, w^u$, obtain the polynomial distribution over the sentiment words according to the hyperparameters of the Dirichlet distribution $\eta^u$.

$$\pi^u \in Dir(\eta^u) \tag{5}$$

At this point, for each document in the travel platform $d^u$, a polynomial distribution can be obtained for each document based on the Dirichlet parameter $a^u$.

$$\psi_a^u \in Dir(a^u) \tag{6}$$

For each text word in the document $d^u$ $w$, the subject $z_a^w \sim Multi(\psi_a^u)$ is obtained according to the distribution $\psi_a^u$; then, get the text word $w \sim Multi(\psi_{z_a^w}^u)$ based on the distribution $\psi_{z_a^w}^u$.

For each visual word in the document $d^u$ $v$, first get the subject $z_a^v \sim Multi(\psi_a^u)$ according to the distribution $\psi_a^u$, then get the text word $v \sim Multi(\psi_{z_a^v}^u)$ based on the distribution $\psi_{z_a^v}^u$.

For each sentiment word in the document $d^u$ $s$, first obtain the topic distribution $\tau \sim Uniform(z_1^u, z_2^u, z_3^u, \ldots, z_k^u)$. Then get the sentiment word $s \sim Multi(\pi^u)$ based on the distribution $\pi_1^u$.

$$\begin{cases} w \sim Multi(\psi_{z_a^w}^u) \\ v \sim Multi(\psi_{z_a^v}^u) \\ s \sim Multi(\pi^u) \end{cases} \tag{7}$$

So far, the three-word senses of Eq. (7) have been obtained. The feature expressions of the tags are constructed by decomposing each tag into spatial vector models. The distribution of conceptual features in images shared by users in different interest categories is different as are the posted travel texts and the user labels, so their interests can be predicted accordingly. Given a certain user U, suppose the set of images contained in his posted information is $S = i_1, i_2, \ldots, i_n, n$ denotes the number of images, and for each image i, the VGG19 model is used to extract image features. The process of partial image feature representation is shown in Fig. 4, and the output of the second fully connected layer (FC2) is selected as the feature vector of the image $V_i$. The fused multiple image feature vectors are averaged and input to the fully connected layer, and the output is used as the feature representation of the image part.

Finally, based on the text part feature $V_t$ and the image part feature $V_{pic}$, the relationship matrix after multi-modal data fusion is obtained as follows:

$$A = V_t \otimes V_{pic} \tag{8}$$

where, $\otimes$ represents the direct product operation. The mean value of matrix A is calculated by column, and the calculation result is input into the 2-layer fully connected layer, and the final judgment result is output.

## EXPERIMENT AND ANALYSIS

### Data processing

This section runs simulation tests to compare the performance of the proposed model in this study with the methods described in the literature to assess the effectiveness of the model (*Javed et al., 2021*; *Park & Kim, 2021*, and *Fedorov et al., 2022*) for comparative analysis. Firstly, the dataset is processed by crawling data from the travel website TripAdvisor, which is then used to construct the dataset for evaluating the comparative

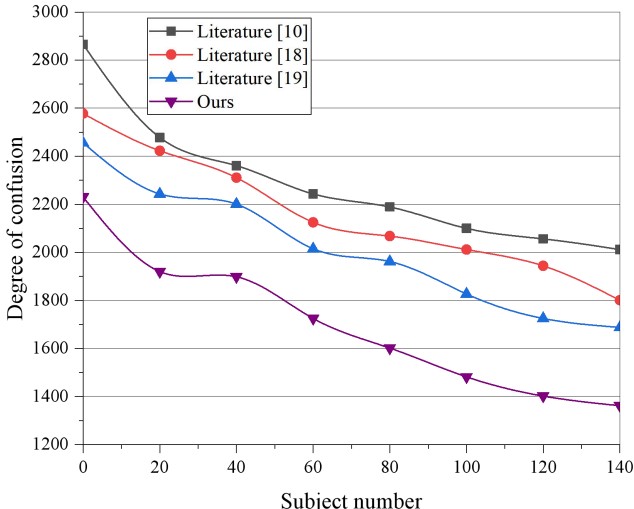

**Figure 4** **The degree of confusion among different models.**

experiments. The multi-modal data, consisting of text and travel images, yields 17,398 tourists' data, including 16,782 text comments and 35,628 images. In order to analyze the text data, text words are extracted. Since nouns are more likely to represent the objective meaning of things and are thus more suitable for describing the properties of things, all nouns in the documents are extracted as text words in this article. Additionally, adjectives, verbs, and adverbs are more likely to carry evaluation or emotional tendencies of things, and are thus used as emotional words for word sense segmentation. To distinguish the word markers in the document into nouns, adjectives, verbs, and adverbs, the lexical annotation function provided by Stanford NLP toolkits is utilized. In terms of image processing, the final visual content of the images is represented by 1,021 visual words. Specifically, the images are first chunked and clustered using the K-means method based on feature extraction, resulting in a final image.

## Comparative performance

The efficacy of the proposed model in the travel recommendation task is evaluated based on a text feature length of 1,000 and image features of 1,000 to verify whether the multi-modal data fusion and mining method proposed in this article can improve the accuracy of interest classification to a significant extent. Moreover, it is verified that the content classification model structure based on image and text fusion is better than the traditional unimodal classification results. The number of fully connected layer nodes in *Park & Kim (2021)* is 4,096, with two fully connected layers, and each training input batch contains 64 training data. The dataset is divided into two parts for quantitative experiments, with 80% set as the training set and the remainder as the test set. Figure 4 displays the variation of each scenario's perplexity scores under different topics. Based on the experimental results, 100 is chosen as the number of iterations for the experiment. It can be observed from the results that *Allan et al. (2003)* achieved the highest perplexity, which implies the

**Table 1  Comparison of F1 and recall rate of different models.**

| Literature | Recall (%) | F1 (%) |
|---|---|---|
| *Allan et al. (2003)* | 81.4 | 81.7 |
| *Park & Kim (2021)* | 83.1 | 83.1 |
| *Fedorov et al. (2022)* | 84.5 | 83.9 |
| Ours | 86.1 | 86.4 |

worst generalization ability, as *Allan et al. (2003)* only models text words and sentiment words without distinguishing between them. *Park & Kim (2021)* and *Fedorov et al. (2022)* improve the generalization ability based on the additional dependency between visual or sentiment information and textual information. However, compared to the models in *Allan et al. (2003)*, *Park & Kim (2021)*, and *Fedorov et al. (2022)*, the proposed model in this article performs better and obtains the best results.

Meanwhile, in order to evaluate the performance of the schemes in terms of F1 and Recall, relevant experiments were compared. Table 1 shows that the F1 and Recall values of the other schemes are all lower than 85%, and the model constructed by the text has the best effect, achieving the best results among all the models, with the F1 value of 86.4% and the recall rate of 86.1%. All indexes are higher than those of other models, indicating that our model is more suitable for the travel recommendation system, and its recall rate and F1 value will be increased to 3%–5%.

To assess the efficacy of the recommendation system, we selected 1,261 tourists who had visited at least 15 attractions after the completion of the model learning. In this study, we measured and evaluated the performance of the proposed algorithm using Precision and MAP following traditional search metrics. The Precision and MAP values for the two experiments, when LOPs are 5, 10, and 20, are presented in Figs. 5 and 6, respectively. *Allan et al. (2003)* performs poorly because it cannot explore the potential relationships between multi-modal topics and emotions. *Park & Kim (2021)* and *Fedorov et al. (2022)* serve better than *Allan et al. (2003)* because they can capture the consistency between different modalities. It indicates that visual or emotional information can enhance recommendation performance. Our proposed model outperforms all experimental comparison models, demonstrating that an effective combination of textual data, visual data, and sentiment can enhance data mining functionality and assist in implementing recommendation applications.

## DISCUSSION

Given that the abundance of multi-modal information characterizes tourism data, the conventional single-modal data mining algorithms for interest recommendation suffer from low accuracy. In order to address this issue, this article proposes a multi-modal emotion-aware topic model that mines features such as text and images in tourism data. By leveraging this model and a travel recommendation algorithm based on multi-modal data mining and multi-vector word sense segmentation, the travel recommendation platform can provide users with personalized tourist attraction recommendations. Unlike

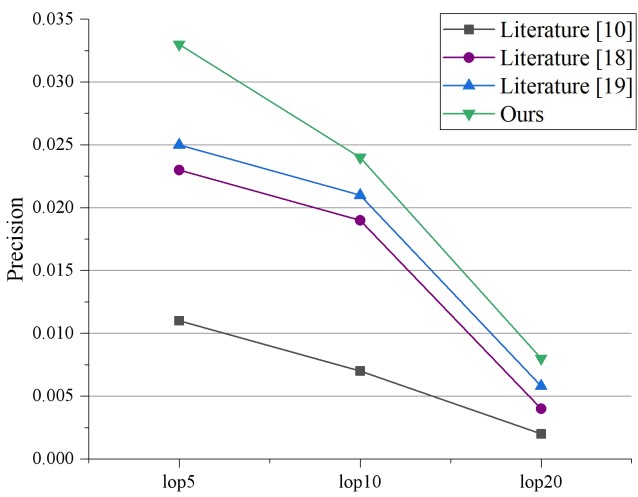

**Figure 5** Precision comparison under different LOPs.

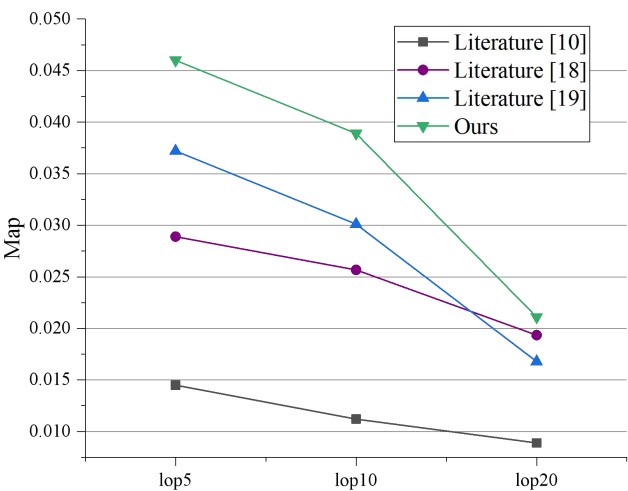

**Figure 6** Comparison of scheme map under different LOPs.

traditional multi-modal recommendation methods that learn and integrate different modal data separately, the proposed scheme considers both "multi-modal data" and "tourism theme" to improve the recommendation efficiency. Simulation experiments demonstrate that the model accurately captures user emotions, resulting in a higher accuracy in tourist attraction recommendations. In the future, consumers in the tourism market will pursue more high-tech and intelligent tourism services. By applying deep learning technology to the travel industry, we can dig deep into users' travel data and figure out their preferences. The implementation of this recommendation algorithm helps the tourism platform to: carry out mining, analysis and reuse according to the user's data information; combine the data and emotion analysis; dig tourism data deeply; provide users with accurate

recommendation of tourist attractions and superb service level; promote the development of the tourism platform; and promote the application of the model, which will change the operation mode of the tourism market and improve the service quality and efficiency of tourism enterprises.

## CONCLUSION

Although numerous outstanding research outcomes have been in recommendation algorithms, the relatively mature outcomes are predominantly focused on e-commerce and related domains. Further enhancing the precision and personalization of recommendations continues to be the prime objective of researchers' endeavours. In this article, we have presented data mining algorithms for recommendation applications from the perspective of actual tourism data. We have established multi-modal data mining for textual and image data in tourism data, employing a theme sentiment model to mine potential themes and sentiments in textual travel data. This method facilitates the effective identification of theme and sentiment words in both text and images, thus enabling multi-vector word sense segmentation. Ultimately, we utilized the Python Scrapy framework to extract authentic data from TripAdvisor, a leading travel social networking site. This process enabled us to build a multi-modal dataset for training travel personalized recommendation algorithms, which could be utilized for model training and testing. The experimental results revealed that the proposed scheme's recommendation efficiency is substantially greater than the traditional multi-modal recommendation method, which learns the data of distinct modalities separately and subsequently integrates the scores. The findings attest to the universality of the proposed model and effectiveness of the recommendation algorithm.

### Funding
The author received no funding for this work. The funders had no role in study design, data collection and analysis, decision to publish, or preparation of the manuscript.

### Competing Interests
The author declare that they have no competing interests.

### Author Contributions
- Ruixiang Liu conceived and designed the experiments, performed the experiments, analyzed the data, performed the computation work, prepared figures and/or tables, authored or reviewed drafts of the article, and approved the final draft.

### Data Availability
The figures and dataset are available in the Supplementary Files.

## Supplemental Information

Supplemental information for this article can be found online at http://dx.doi.org/10.7717/peerj-cs.1436#supplemental-information.

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
