# Peer review of "Development of a travel recommendation algorithm based on multi-modal and multi-vector data mining"

_PeerJ Computer Science, doi:10.7717/peerj-cs.1436_

## Round 0.1 · original submission · Minor Revisions

Dear author,

Your article has a few issues. We encourage you to address the concerns and criticisms of the reviewers and resubmit your article once you have updated it accordingly.

Best wishes,

·

Basic reporting

There are several grammatical problems within the text.
Furthermore, all figures require high resolutions printings. Also, check the text and fonts in Figures.

Experimental design

Methodology is not well designed. For readers to quickly catch your contribution, it would be better to highlight major difficulties and challenges, and your original achievements to overcome them, in a clearer way in the abstract and introduction. Also, the methodology section is also not well presented.

Validity of the findings

I feel that authors have some how contributed through their proposed study in the proposed area of trave recommendations.

Additional comments

the manuscripts should be recommended for publication after minor revision with grammatical mistakes, and methodology design.

Reviewer 2 ·

Basic reporting

In order to effectively utilize the highest potential value of tourism data available on the Internet, that is, multi-modal data providing larger capacity and richer content on social media, this paper proposes a new travel recommendation algorithm based on multi-modal data mining, which provides greater economic and social value for the development of tourism. Overall the paper seems to be written well and having some novel contributions , however, it can be improved further by the following suggestions:

(1) In the abstract part of this paper, it is suggested to use experimental data to present the results of the algorithm.
(2) The references related to the work in section 2 of the paper are superscripts, while the references in sections 3 and 4 are not superscripts, such as [2] and [2]. Please check and unify the format.
(3) Formulas (1),(2) and (3) on page 5 of the paper all use Dirichlet function, but the definition and function of the function are not explained in the paper, so it is suggested to supplement.
(4) Why does this paper adopt the multivector semantic segmentation technique? Tell me the advantages of this technique.
(5) In Section 4.2, 1261 tourists are selected to evaluate the system, but the data amount is too small.
(6) There are too few relevant experimental contents in this paper, so it is suggested to add relevant model evaluation indexes, such as R, F1, etc.
(7) Formula (8) in Section 3 of this paper fuses text features and image features for output. Is it classification or regression? What is the output result? Please elaborate.
(8) In the conclusion part of the paper, it is suggested to increase the concrete achievements, advantages and future development prospects of the tourism recommendation model.

Experimental design

no comment

Validity of the findings

no comment

Additional comments

no comment

---

## Round 0.2 · accepted · Accept

Dear Authors,

Thank you for the revision. Your paper was accepted following the second peer review process.

Best wishes,

·

Basic reporting

The structure/formatting are appropriate, and the paper is well-organized. The abstract, introduction, methodology, results, and references sections are all well-written and meet the required standards for publication.

Experimental design

The experimental design is also well-designed.

Validity of the findings

The findings of the study are valid and reliable. The results are presented clearly. Overall, the study's findings contribute to the existing knowledge in the field.

Additional comments

I have reviewed the revised version of the paper, and I am pleased to recommend it for publication. The authors have taken into account the comments and suggestions provided and have made appropriate revisions. Based on their efforts, I believe this paper is now suitable for publication.

Reviewer 2 ·

Basic reporting

Authors has done requested revision

Experimental design

no comment

Validity of the findings

no comment